# Autonomous Enterprise as a Model of Hotel Operation in the Aftermath of the COVID-19 Pandemic

Małgorzata Sztorc 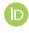

Department of Management and Marketing, Faculty of Management and Computer Modelling, Kielce University of Technology, 25-314 Kielce, Poland; msztorc@tu.kielce.pl; Tel.: +48-41-34-24-309

**Abstract:** The article aims to identify modern technological solutions in the field of automation and robotization of business processes that change the way hotel enterprises operate in the face of the global COVID-19 pandemic. Therefore, the research problem was expressed in the form of a question: what tools favor the dynamic digital transformation of hotel functioning models and the implementation of the paradigm of intelligent and autonomous entities operating in the conditions of the COVID-19 pandemic on the hotel market? During the empirical research, the use and implementation of new solutions in the field of innovative technologies supporting the optimization of processes taking place in hotel enterprises were considered. In addition, technologies and tools were identified that are considered strategic from the perspective of implementing the concept of an autonomous hotel carried out under the influence of the crisis related to the COVID-19 pandemic. Due to the specific purpose of the article, a survey was conducted using the questionnaire method with the Computer Assisted Web Interview technique alongside a self-developed questionnaire. The survey was carried out in June–September 2021 among 462 representatives of hotel companies operating in the three-, four- and five-star standards. The results of the diagnostic survey were statistically analyzed using the Statistica and RStudio software packages. To interpret the obtained data, the descriptive method, principal component analysis, and cluster analysis were used. As a result of the conducted research, it was noticed that the COVID-19 pandemic obligated hotels to automate processes that made it possible to connect devices within digital ecosystems and to optimize processes inside vertical and horizontal value chains. Such activities enabled the creation of a new model of hotel functioning referred to as an autonomous enterprise. Nowadays, one of the most important strategic behaviors of hotel entities is the ability to quickly counteract crises resulting from the COVID-19 pandemic and the related choice of the right course of action, as well as the ability to immediately make the right decisions. The research results may motivate the implementation of new technologies in the area of transforming and developing digital business models by hotels. The issues presented in the article are an attempt to fill the gap by pointing out practical experiences related to the use of individual technological solutions and their effectiveness in process automation and implementation of autonomous models of functioning of entities in the hotel services sector.

**Keywords:** BPA; RPA; business model; autonomy; hotel industry; COVID-19

## 1. Introduction

In the modern market, we can observe exceptionally intensive technological development determined by full automation, robotization, and digitization of processes carried out in enterprises. In addition, the activities of entities on regional and global markets in the conditions of the new reality resulting from the COVID-19 pandemic justified the importance of implementing modern tools and technologies for autonomy. Their multifaceted application influences profound transformations in the operation of enterprises and the hotel services sector [1]. Such changes affect the transformation of the operating models of hotel facilities, their services, processes, and organizational structures as a result of using advanced technologies, as well as automation and robotization tools [2].

The literature on the subject recognizes the role of automation and robotization of business processes [3–6] but there are still unexplored issues related to the answer to the key question: what tools are conducive to the dynamic transformation of hotel functioning models and the implementation of the paradigm of intelligent and autonomous entities operating in the conditions of the COVID-19 pandemic in the hotel market? In connection with the above, the subject of autonomous enterprises is an important issue from the perspective of the analysis of tools responsible for creating a new model for the functioning of hotel enterprises. For this reason, the article attempts to identify modern technological solutions in the field of automation and robotization of business processes that change the way hotel enterprises operate in the face of the global COVID-19 pandemic.

Due to the aim and research question, as well as the analysis of the literature on the subject in the scope of the analyzed issues, a research hypothesis was formulated, which indicates that hotel enterprises, through the processes of automation, robotization, and optimization, implement modern tools and technologies shaping the new model market operation based on the autonomization of functions and tasks that create value for hotel guests in the turbulent reality caused, among others, by the COVID-19 pandemic. Therefore, a survey was conducted using the questionnaire method alongside the Computer Assisted Web Interview (CAWI) technique with the use of a self-developed questionnaire. The source material for the empirical study was obtained during the survey, carried out in June–September 2021 among 462 representatives of hotel enterprises operating in the three-, four- and five-star standards with Polish and foreign capital on the Polish market. To interpret the obtained results, statistical tests (including chi-square) and multivariate methods of statistical data analysis (principal component analysis, cluster analysis) were used, which were verified using the Statistica and RStudio software packages.

The issues analyzed in the article have not been the subject of research for the hotel market so far. For this reason, it intends to supplement the information in the research carried out so far on practical experiences related to the automation and robotization of processes inside enterprises.

*Literature Review*

Initially, the digital transformation of the organization was aimed at ensuring the highest quality of hotel guest service [7]. A symptom of digitization is the continuous automation of business processes (BPA-Business Process Automation), in particular, their robotization with the use of IT tools included in the RPA (Robotic Process Automation) category. For this reason, special attention is currently being paid to the automatic implementation of operational processes and support for entities, which does not provide direct value to customers [8].

The first stage of implementing automation solutions was the use of chatbots for customer service, which redirected customer calls to the appropriate operators. Then, such solutions were gradually replaced by software using robotic agents (RPA) capable of automating simple tasks as part of improving the hotel business process (e.g., filling out forms and data mining). RPA tools contributed to the creation of BPA technology, which, in addition to automating tasks, streamlines the decision-making process, data management, content digitization, and automates workflows to improve the efficiency of the entire organization [9].

BPA is a technology-based process of automating tasks performed in the enterprise, the purpose of which is to improve, simplify, reduce costs and increase efficiency in the process of providing hotel services. In addition, it concerns an increasing scope of work while reducing the number of people who perform given tasks using only computer software.

RPA, on the other hand, is a process that engages in performing complex tasks using software, robots, and artificial intelligence (ALI) technology, including Optical Character Recognition (OCR) systems and decision-making models to automate business tasks and processes. Automation concerns routine, repetitive activities performed as individual tasks or entire processes [10].

The goal of BPA and RPA tools is to increase the effectiveness of business functions performed using automation. The former streamlines workflows to improve the performance of a hotel enterprise without focusing on a particular department. Thus, it enables the implementation of software systems that integrate all existing applications, improving the overall specific process. On the other hand, the second tool (RPA) uses software robots (so-called bots) that imitate the activities performed by employees. Consequently, they perform rule-based tasks through user workflow registration [11].

Related to BPA and RPA is a business model called autonomization, which is based on technologies for digitizing processes carried out in hotels. Autonomization means a significant reduction or replacement of employees' tasks through the tasks performed by the work of a robot, machine, or software operating on the principles of self-regulation, which performs specific activities without human intervention [12].

As a result of the global COVID-19 pandemic, hotel companies are implementing Hyper-Automation solutions, which consist of technologies and tools that automate business processes. Thus, it is an element of the digital transformation strategy of the business. The hyper-automation tools include RPA, BPR, Chatbots and Vivebots (tools for automating communication with hotel guests via the website and Call Contact Center systems), Intelligent Business Process Management Systems (iBPMS), Process and Data Mining for Big Data (business process mining tools), Optical Character Recognition (OCR-text recognition tools), LowCode (systems for building applications without coding or with little code) and Artificial Intelligence (ALI) [13]. The literature on the subject indicates that the COVID-19 pandemic caused by the SARS-CoV-2 coronavirus was initiated in the Chinese city of Wuhan at the end of 2019 [14,15].

The pandemic stage is an extremely difficult time for organizations because the regulations of the managing states exclude other sectors of the economy from functioning. As a consequence of the closure of the borders, there has been a decline in international economic exchange and demand for hotel services, which have slowed down. As a result of this type of situation, not only states, but also all hotel enterprises should have a strategic model of operating in crises [16,17]. One of the solutions is the use of technologies based on digitization in the implementation of tasks and processes carried out in hotels, including developed telephone networks, the Internet, and mobile devices. As a consequence, the ideal solution may be the changes made in the business model of hotel entities, conditioning the transition from the digital model to the so-called automating with the use of tools based on BPA and RPA solutions. In the autonomous business model, the range of values for hotel guests and the channels by which the company provides this value are also changing.

## 2. Materials and Methods

The subject of the analysis of this article was to identify modern solutions and technological tools in the field of automation and robotization of processes that change the model of hotel operation from the perspective of the global COVID-19 pandemic. Based on the data obtained during the empirical research, the tools supporting the autonomization in hotel enterprises have been distinguished. The conducted research made it possible to determine four research questions listed in Table 1.

**Table 1.** Research question.

| No. | Research Question (RQ) |
| --- | --- |
| RQ 1 | What intangible and technological resources in terms of equipment with mobile devices and access to the selected type of internet connection are at the hotel? |
| RQ 2 | What types of automation and robotization tools used by hotels affect the improvement of processes in the aftermath of the COVID-19 pandemic? |
| RQ 3 | In what areas of hotel operation are autonomization tools implemented? |
| RQ 4 | What motives and rationale were the hotel managers guided by when implementing the tools of autonomy? |

Source: Own study.

In addition to the indicated research questions in the form of a supplement, the research problem was also included in the form of a hypothesis.

Regarding the above-mentioned issues, a research hypothesis was formulated, which assumes that hotel enterprises, through the processes of automation, robotization, and optimization, implement modern tools and technologies shaping a new model of market operation based on the autonomization of functions and tasks that create value for hotel guests in the turbulent reality caused, among others, by the COVID-19 pandemic. This was verified using the analysis of the obtained research results analyzed in Section 3. The research questions and the hypotheses posed required the use of statistical tests to confirm the existence of significant relationships between the variables obtained during the study. In addition, to verify them, multi-dimensional statistical methods were used in the analysis of the results, such as principal component analysis (PCA) and cluster analysis (CA).

The defined research goal was attempted to be achieved by processing empirical data collected during the quantitative research. Answers to the above-mentioned research questions were obtained using the proprietary questionnaire, which consisted of 28 questions. The survey was carried out using the CAWI (Computer Assisted Web Interview) technique in June–September 2021 among 462 representatives of hotel enterprises (owners, directors, or managers-re-respondents) operating in the three-, four-, and five-star standards with Polish and/or foreign capital. Only one respondent from a hotel of a given category participated in the study and declared the implementation of automation and robotization tools to improve the processes carried out. For this reason, the responses of the respondents were varied and determined by the scope of their substantive and empirical knowledge. When assessing the correctness and completeness of the received questionnaires, 273 (59.09%) forms were qualified for further analysis. In connection with the quantitative verification of the relationships between the variables and the purpose of testing the hypotheses, the chi-square statistic was used. The relationships between the variables are presented in the form of the number and percentage structure of statements received for the questions asked in the compiled categories of three-, four- and five-star hotels (contingency tables). The consequence of this type of statistical test is the test probability (p), the low values of which indicate the statistical significance of the analyzed relationship. The value of the function was calculated using the formula [18]:

$$x^2 = \sum_{i=1}^{l} \sum_{j=1}^{k} \frac{(n_{ij} - \widehat{n}_{ij})}{\widehat{n}_{ij}} = \sum_{i=1}^{l} \sum_{j=1}^{k} \left( \frac{n_{ij}^2}{n_{ij}} \right) - n; \widehat{n}_{ij} = \frac{n_i \, x \, n_j}{n} \tag{1}$$

where: $\tilde{n}_{ij}$ are theoretical numbers, $n_i$—number of sample elements corresponding to the $x_i$ level and the feature $X$, $n_j$—number of sample elements corresponding to the $y_j$ level of the feature $Y$, $n_{ij}$-number of sample elements corresponding to the $x_i$ level and the feature $X$ level and the $y_j$ level of the feature $Y$, $n$—a sum of the sample elements corresponding to the $x_i$ level and the feature $X$ and the $y_j$ level of the feature $Y$, $k$—number of columns, $l$—number of lines.

During the next stage of analysis, the aim of which was to identify tools for the autonomy of processes implemented in hotel enterprises used due to the improvement of processes from the perspective of the COVID-19 pandemic, principal component analysis (PCA) was used. This type of method was used to reduce dimensionality, as well as to visualize the similarities and/or differences between the tested automation and robotization tools used in hotels. PCA reduces the investigated determinants, which remain mutually correlated to a varying degree, to a smaller number of new synthetic variables (so-called principal components-PC) which are collectively uncorrelated, constituting a combination of linear primary variables characterized by the greatest possible variability [19,20]. The method is based on the orthogonal transformation of the input features that are included in a new set with features that are not correlated with each other following the formula [21]:

$$Z_j = b_{j1} \, S_1 + b_{j2} \, S_2 + b_{j3} S_3 + \dots + b_{jn} S_n \tag{2}$$

where: $Z_j$—$j$-th variable ($j$ = 1, 2, ... , $n$), $S_1$ ... $S_n$—main components, $b_{j1}$ ... $b_{jn}$—main component coefficients.

PC consists of linear functions of primary variables which contain, in descending series, information on the structure of data variability. Their number is identified by the scree plot and the Kaiser criterion, for factors with values greater than 1. For the needs of the research process, the factors were rotated by the varimax standard method. On the other hand, for individual factors, variables with maximum factor loadings to a given factor (according to the value of $\geq 0.7$) were determined. The graph of coordinates created shows the relationship between the objects for the principal components. On the other hand, the values of factor loadings remain the correlation coefficients that occur between the obtained data concerning the main components [22].

A multidimensional exploratory technique-cluster analysis (CA)-was used to distinguish areas and identify the motives for the application of autonomization tools by hotel managers. It concerns the problem of grouping variables into disjoint, homogeneous, internal clusters that differ to the greatest extent among themselves [23]. Properly conducted CA enables the classification of a data set into groups due to a more accurate recognition of the information contained in them and recognition of the specificity of identical groups of objects and their comprehensive characteristics [24]. In the conducted research, the agglomeration hierarchical method of ordering and classification was used to distinguish convergent clusters, analyzed due to many statistical features. To determine the distance between the objects, the Euclidean metric was used with the formula [25]:

$$d(x_i , x_k) = \sqrt{\sum_{j=1}^{m} \left( x_{ij} - x_{jk} \right)^2} \tag{3}$$

where: $d$—Euclidean distance between objects, $x_{ij}$—denotes the value of the $j$-th variable for the $i$-th object, $m$—number of all diagnostic variables.

On the other hand, the systematization of objects into clusters was made using the unweighted pair group method (UPGMA-unweighted pair-group method using arithmetic averages), which is based on determining the distance between two clusters using the arithmetic mean determined from the distance inside all pairs of objects belonging to identical clusters according to the pattern [26]:

$$d(C_i, C_j) = \frac{1}{n_{C_i} n_{C_j}} \sum_{i \in C_j} \sum_{j \in C_j} d(x_i, x_j) \tag{4}$$

where: $n_{C_i}$, $n_{C_j}$—are the numbers of objects in the cluster $C_i$, $C_j$, $d$ ($x_i$, $x_j$)—Euclidean distance, respectively.

The hierarchical cluster structure formed during the analysis was presented in the form of a dendrogram (cluster trees).

Data analysis in the field of quantitative research was carried out with the use of the Statistica and RStudio statistical packages.

*Characteristics of the Research Sample*

After the rejection of incorrectly completed forms, 273 hotel companies operating on the Polish and foreign markets participated in the study. The selection of the research sample was deliberate, non-random, and depended on the following criteria: (1) enterprise from the accommodation services sector; (2) hotel operating in the three- (3 *), four- (4 *) or five-star (5 *) standard; (3) a facility operating on the market for over seven years; (4) an enterprise with the following categories: small, medium or large; (5) an entity with domestic or foreign capital.

The survey was carried out among 53% of three-star hotels (including 38% with Polish capital [hp] and 15% with foreign capital [hf]), 38% four-star facilities (including 15% hp and 23% hf) and 9% five-star (including 3% hp and 6% hf).

The hotel companies that participated in the study were also distinguished by their form of ownership. Private entities were the most numerous in their group, constituting 78% of the total number of participants in the study. This group was dominated by Polish capital (hp-56%) and foreign capital constituted 44%. The headquarters of the facilities were in Poland—91 hotels, France—59 facilities, the United States—47 entities, England—44 companies, and 32 in Sweden.

Hotels operating on the market for over 15 years constituted a significant group-64%. In turn, 28% of enterprises have been operating for 10–15 years, and the remaining examined objects (8%) for less than nine years. The stage of the market activity of hotels makes it possible to obtain knowledge from the perspective of understanding the opinions of respondents expressed in the survey, which are based on long-term experience in the implementation of process automation and robotization tools. Medium-sized enterprises dominated the studied sample and constituted 73% of hotels. On the other hand, 16% of the group consisted of small enterprises, while large enterprises were represented by 11% of entities. The respondent's group consisted of persons holding managerial positions: owners (25%), directors (41%), and managers: department managers (34%) in the surveyed hotels.

Due to the complexity of the research and its international character, its cognitive potential should be emphasized.

## 3. Results

One of the basic goals of operating hotel enterprises is the effective management of their resources. Nowadays, as a result of the pandemic crisis, the hotel services sector is undergoing a rapid transformation. This is the result of the implementation of modern technologies and tools that systematically improve and optimize this type of activity.

### 3.1. Internet Connectivity Technologies and Mobile Devices in the Process of Hotel Autonomization

Based on the research, in the first stage, the abundance of mobile devices and access to the selected type of internet connection in hotel enterprises were determined. This type of potential is a consequence of the development of automation and robotization tools through the popularization of the use of the Internet and mobile devices as modern information carriers, as well as innovative methods of communication between hotel facilities, the market, and buyers. The purpose of using modern 5G technologies is to increase the efficiency and effectiveness of hotel management and to shape a new model for the functioning of enterprises in the conditions of the COVID-19 pandemic.

The basic instrument that modifies the existing methods of hotel business operation and, at the same time, enables the development of new ones is the Internet and related mobile technologies. The research shows that hotels mainly use the following types of internet: fiber-optic (92%-5\*hotels, 75%-4\*h, 79%-3\*h), 4G (71%-5\*h, 62%-4\*h, 39%-3\*h), 5G (40%-5\*h, 38%-4\*h, 31%-3\*h), 3G (4%-5\*h, 67%-4\*h, 21%-3\*h), and radio (4%-5\*h, 39%-4\*h, 41%-3\*h). In turn, an internal computer network is available to almost every five-star hotel (98%), 94% of four-star entities, and less than half of three-star facilities participating in the study (44%). In addition, respondents indicated that hotel enterprises use Bluetooth wireless data transfer technology (86%-5\*h, 90%-4\*h, 24%-3\*h) and Beacon devices based on this solution (38%-5\*h, 38%-4\*h, 6%-3\*h). In turn, equipment with mobile devices, such as notebooks, was declared by all respondents (100%) from five- and four-star hotels and 41% from three-star hotels. Most hotels also have multimedia devices, such as a smartphone (100%-5\*h, 97%-4\*h, 91%-3\*h) and a tablet (69%-5\*h, 75%-4\*h, 16%-3\*h). Small netbook computers are used by 48% of 5\* hotels, 61%-4\*h, and only 12%-3\*h.

Access to modern technologies is the criterion for the functioning of hotel facilities. Nowadays, most of them during process automation use various types of technologies based on the Internet to change or modify the model of functioning towards autonomy. Technologies of this type are treated as a generator intensifying the development and digital transformation of hotel enterprises. Therefore, using the independence test $\chi^2$ (see Table 2), the following hypotheses were verified:

**Hypothesis 1 (H1):** *The use of the selected internet variant and the type of mobile device does not depend on the standard of the hotel.*

**Hypothesis 2 (H2):** *The use of the selected internet variant and the type of mobile device depends on the standard of the hotel.*

**Table 2.** Results of the test statistics for the internet variant and the type of mobile devices used in hotels.

| Statistics | $\chi^2$ | df | *p* |
|---|---|---|---|
| Pearson's Chi$^2$ | 165.219 | df = 2 | *p* < 0.001 |
| Chi$^2$ LR | 164.867 | df = 2 | *p* < 0.001 |
| Fi | 2.1423 | | |
| Contingency coefficient | 0.90614 | | |
| Cramer V | 0.9984 | | |
| | $\chi^2 = 165.219, p < 0.001$ *** | | |

where: the adopted level of significance, difference $\alpha = 0.05$; the number of degrees of freedom df = 2; the critical value of the chi-2 test for df = 2 according to the tables is 5.991; *** statistically very significant probability. Source: Own study based on the RStudio package.

The analysis of the results of the Pearson $\chi^2$ test of independence shows that there is dependence among the examined variables. The $\chi^2$ statistics was 165.219 and with two degrees of freedom, while the critical value of $\chi_\alpha^2$ is 26.296. On this basis, the null hypothesis should be rejected, and an alternative hypothesis should be adopted, which indicates the relationship between the use of a specific type of internet and mobile devices and the hotel standard. The level of dependence calculated based on the value of the contingency coefficient (0.90614) and the VCramer coefficient (0.9984) should be considered strong. Moreover, $\chi^2$ and the VCramer dependence showed the statistical significance of the studied correlation at the level of the 1st type error, which is 1%.

According to the analysis of the data obtained in the first stage of the research, the growing demand for access to mobile devices and modern technologies based on the use of 5G Internet in hotel enterprises should be emphasized. This tendency is a response to the dynamic development of automation and robotization tools and the related new model of functioning. It results from the autonomization, robotization, and adaptation of hotel enterprises to the contemporary market conditions in the reality of the COVID-19 pandemic.

*3.2. Tools for the Autonomization of Hotel Enterprises*

The functioning of hotel enterprises in the current market conditions is related to the implementation of modern technologies that improve business processes and cyclical tasks performed by employees. In hotel development strategies, a strategic goal is a digital transformation, which is being influenced by the COVID-19 pandemic. For this reason, the first stage of the research aimed to identify tools influencing the autonomy and robotization of business processes in hotels. The answers provided by the respondents were interpreted using PCA. Among them, there are tools used in enterprises that condition their autonomous development.

Based on the obtained data, a PCA projection was prepared, from which it should be concluded that, depending on the category, hotel enterprises use various types of tools helpful in the process of automation and robotization of processes, affecting the autonomous model of functioning. The relationships between the original data and the obtained principal components are presented in Figure 1. During the visual projection of PCA, the interrelationships and disproportions between the analyzed variables in the system of the first two principal components were exposed.

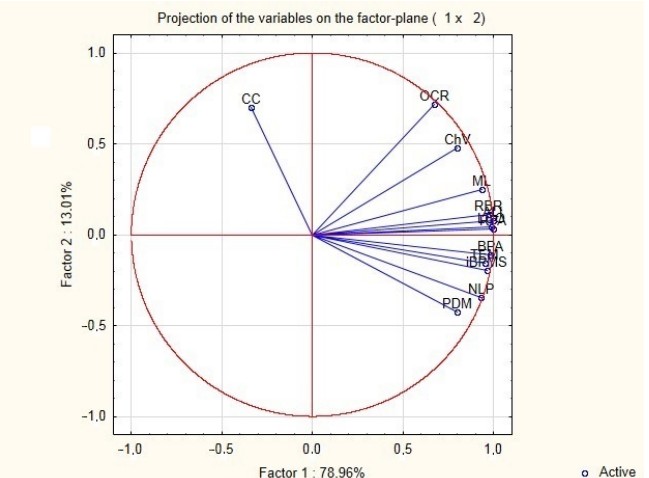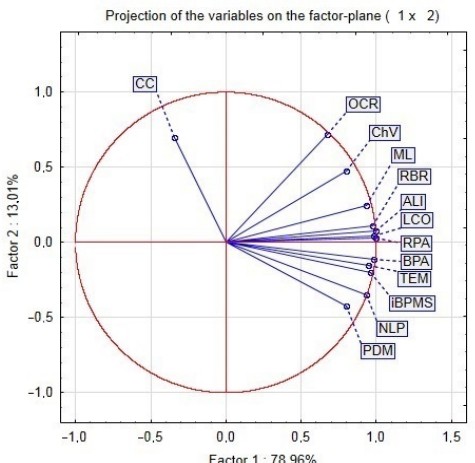

**Figure 1.** Variables of the applied tools influence the autonomy of hotel enterprises in the two-dimensional space of the first and second components. Where: RPA—Robotic Process Automation; ChV—Chatbots Voicebots; iBPMS—Intelligent Business Process Management Systems; PDM—Process and Data Mining for Big Data; OCR—Optical Character Recognition; LCO—LowCode; ALI—Artificial Intelligence; RBR—Rule-Based Robotics; TEM—Text Mining; NLP—Natural Language Processing; ML—Machine Learning; CC—Cognitive Computing; BPA—Business Process Automation. Source: Own study based on the Statistica 13.3 package.

A significant part of the analyzed data is of key importance, which is justified by the range of the vectors. Given that they are elongated, and the variables located near the circle, it should be concluded that the predominant components of the information contained in the input data are transferred by the main components.

The performed analysis justifies that the decisive factor is the first principal component, which explains 78.96% and the second one interprets 13.01%, therefore they simultaneously explain 91.97% of the variance of the original variables. Based on the obtained results, it should be concluded that factor loadings with a positive correlation are responsible for the application of the following tools influencing the autonomy of hotel enterprises: Rule-Based Robotics (RBR) and Artificial Intelligence (ALI), LowCode (LCO)-Robotic Process Automation (RPA), Business Process Automation (BPA)-Text Mining (TEM), Intelligent Business Process Management Systems (iBPMS)-Text Mining (TEM). On the other hand, among the negatively correlated factors, one should mention hotels that use the following autonomization tools: Cognitive Computing (CC) and Process and Data Mining for Big Data (PDM). In addition, the analyzed variables are also characterized by uncorrelated determinants, which include: Process and Data Mining for Big Data (PDM)-Chatbots and Voicebots (ChV), Natural Language Processing (NLP)-Optical Character Recognition (OCR), Intelligent Business Process Management Systems (iBPMS)-Machine Learning (ML) and Robotic Process Automation (RPA)-Process and Data Mining for Big Data (PDM). During the next stage of the analysis, based on the Kaiser criterion [27] and the Catell chart, an adequate number of factor loadings was selected. In line with the assumptions of the PCA method, the first four factors with eigenvalues above 1 should be adopted for the next stage of the study. The selected categories justify a total of 99.47% of the variance of all 13 analyzed variables (see Figure 2a). On the other hand, the factorial screen is at the level of the fifth factor. Therefore, four factors should be taken for further analysis, because on the right side of factor 4, which has the value of 1.63%, there is a slight decrease in eigenvalues. Thus, according to the conducted analysis, it is possible to transform 13 variables into five orthogonal components.

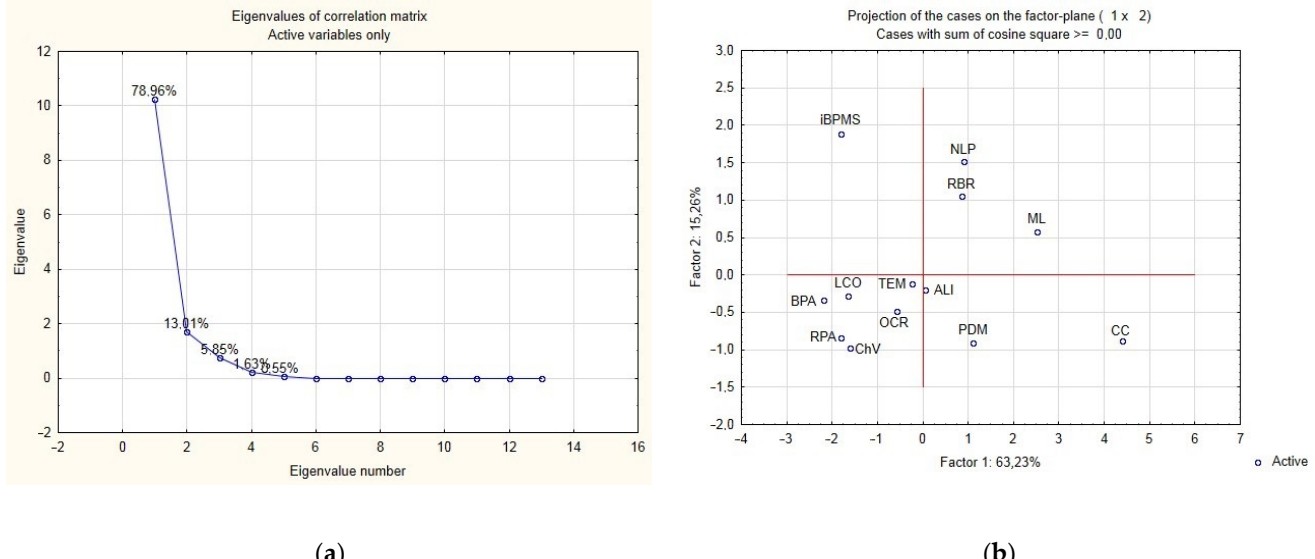

**Figure 2.** Principal Component Analysis; (**a**) Cinder site showing the contribution of each component to the explanation of the overall variance; (**b**) a scatter plot of the autonomization tools used by hotels in the space of the first two main components. Where: data are the same as in the description of Figure 1. Source: Own study based on the Statistica 13.3 package.

The consequence of PCA is the formation of a new set of orthogonal variables and the determination of the coordinates that make up this type of system, the so-called factorial cases. With the use of points with coordinates, a scatter plot was created for the first two principal components (see Figure 2b). The linear map formed as a result of the projection process shows three uniform groupings of points, which inform the following structure of tools used in the hotel autonomization process: (1) LowCode (LCO); Text Mining (TEM); Business Process Automation (BPA); Robotic Process Automation (RPA); Chatbots and Voicebots (ChV); Optical Character Recognition (OCR); (2) Artificial Intelligence (ALI); Process and Data Mining for Big Data (PDM); (3) Natural Language Processing (NLP); Rule-Based Robotics (RBR); Machine Learning (ML). Nevertheless, the chart also shows the tools for the autonomy of hotel enterprises, which differ significantly from those selected above due to the two variables included in the analyses. The first main component has a decisive influence on their distribution. As a result, Intelligent Business Process Management Systems (iBPMS) and Cognitive Computing (CC) are the tools that stand out from those indicated in the three groups.

The modern conditions of the functioning of hotel enterprises in the conditions of the COVID-19 pandemic encourage managers to implement autonomization tools that, in terms of automation, improve the work of staff. Through automation, hotels that are skillfully able to use digital tools and technologies by transforming processes and business models minimize the dependence on employee interference in cyclical tasks.

Based on the conducted research, it should be concluded that the most popular autonomization tools used by hotel enterprises with Polish capital (hp) include RPA, LCO (87.5%-5*hp), iBPMS (89.04%-4*hp and 57.3%-3*hp), ChV (75.3%-4*hp) and BPA (48.9%-3*hp). In turn, in hotels with foreign capital (hf), the autonomization process is carried out using the following tools: BPA (97.2%-5*hf, 93.8%-4*hf, 94.7%-3*hf), OCR (94.4%-5*hf, 96.9%-4*hf), ChV (94.4%-5*hf, 89.5%-3*hf), RPA (88.9%-5*hf, 92.3%-4*hf, 84.2%-3*hf) and iBPMS (98.5%-4*hf).

On the other hand, the least popular tools in hotels with Polish capital have so far been the autonomy tools in the field of: ML (18.8%-5*hp, 27.4%-4*hp, 20.8%-3*hp), OCR (31.3%-5*hp, 17.4%-3*hp), RBR (31.3%-5*hp), CC (31.3%-5*hp, 5.5%-4*hp, 1.1%-3*hp) and PDM (35.6%-4*hp). The following tools are sporadically used in hotels with foreign capi-

tal: RBR (52.7%-5*hf), CC (52.7%-5*hf, 15.4%-4*hf, 36.8–3*hf), PDM (33.8%-4*hf) and ML (39.5%-3*hf).

According to the results of the obtained research, it should be concluded that optimally applied tools of autonomization in hotels should contribute to a significant increase in the effectiveness and efficiency of activities in the process of providing services and the market model of functioning.

### 3.3. Areas of Application of Autonomization Tools in Hotels

The next stage of the research concerned the domains of hotel business operation, in which automation and robotization tools are used to autonomize business processes.

The turbulent environment related to the COVID-19 pandemic, in which organizations operate today, creates the need to support business processes. As a result of the indicated circumstances, hotel enterprises are obliged to introduce changes in the nature of innovation and transformation in all areas of business activity. Modern tools improve the implementation of tasks and enable the hotel and the buyer of services to co-create value.

The concept of generating and retaining value is a key component that constitutes the hotel operating model. Thus, the fundamental function of development in hotel enterprises is performed by advanced solutions within the framework of autonomization tools and information and communication technologies (ICT) in the field of supporting business processes and streamlining the decision-making process. As a result, hotels use integrated IT systems, including those in the fields of hotel services, hotel guest service, reservation system, and supporting the hotel facility management process. Thanks to the robotization of processes, they enable the organization of endogenous company processes, processing, storage, recording, data selection, and cooperation with strategic contractors and buyers. The progressive intensification of the automation process and modern management methods resulted in the formation of ERP (Enterprise Resource Planning) and MRP (Manufacturing Resource Planning) systems, which provide support for most areas of hotel business operation.

The collected empirical data made it possible to identify areas in which hotel enterprises use ICT-based tools and systems to support business processes, shaping a new model of their operation. To group the analyzed areas of operation of hotel enterprises in which modern solutions are implemented, cluster analysis was used according to the classification of average connections with the use of the Euclidean distance (see Figure 3).

Based on the obtained data, homogeneous subgroups with the same patterns were distinguished, as was a group characterized by different features, which was classified into different subsets. The number of clusters was selected based on the agglomeration diagram (see Figure 3a). The process of grouping the areas of application of autonomization tools due to the similarities in the use of modern automation and robotization systems shaping a new model of hotel enterprises functioning in the conditions of the COVID-19 pandemic was stopped at the 9th stage of agglomeration, which corresponds to a distance of 30. Moreover, this was also the case in the last 13th step with a distance of up to 81.5.

As a consequence of grouping the areas of supporting business processes similar in terms of the adopted diagnostic features describing the level of technology use (see Figure 3b), five clusters should be distinguished. On their basis, the following groups were formed: one- (one cluster), two- (two clusters), four- (one cluster), and five-element (one cluster). The first with the highest level of use of automation and robotization tools to support business processes covers four areas: hotel guest service (HS), online booking support (SP), reception support systems (RW), and online sales channel management (CS). The second distinguished cluster consists of two areas in which the processes are supported: service management (SM) and customer relationship management (RM). In turn, the third domain is represented by a five-element focus that combines the following variables: document flow automation (AD), content management (CM), office work support (OW), online distribution channel management (DC), and supplier relations management (MC). On the other hand, the sphere of finance and accounting has been classified as the fourth

separate single-element cluster. The fifth group covers the areas of servicing purchasing processes (PP) and human resources (HR), which are characterized by a low level of use of tools supporting business processes.

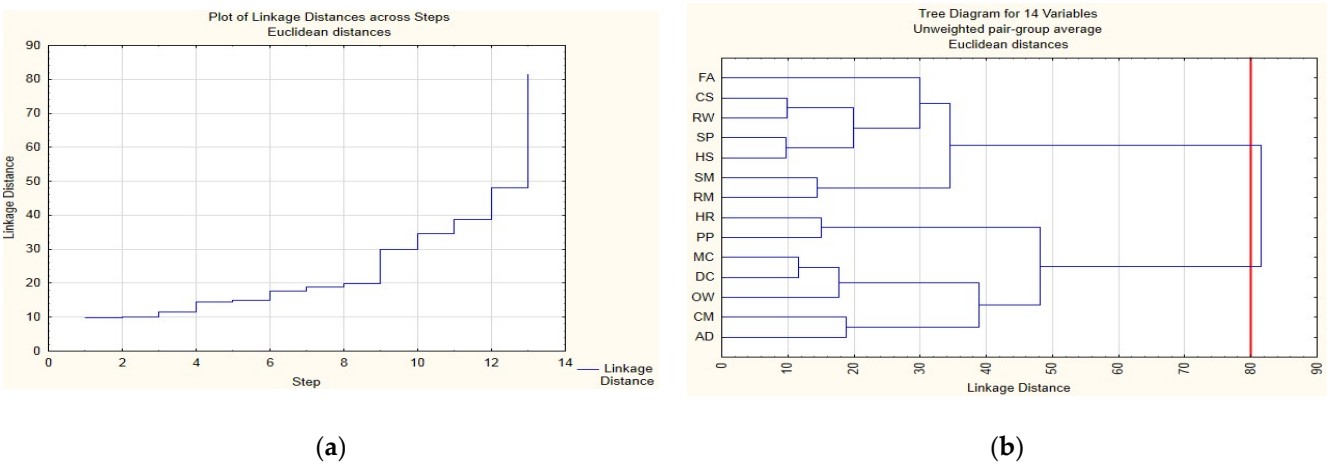

**Figure 3.** Hierarchical cluster analysis for business process support areas; (**a**) agglomeration diagram (**b**) dendrogram showing the areas of ICT use, separated based on cluster analysis. Where: FA-finance and accounting; HR-human resources; SM-service management; RM-customer relationship management; MC-managing contacts with suppliers; CS-online sales channel management; DC-managing online distribution channels; CM-content management; OW-supporting office work; RW-supporting the reception work; PP-handling purchasing processes; SP-online booking service; HS-hotel guest service; AD-document flow automation. Source: Own study based on the Statistica 13.3 package.

The conducted research shows that the management processes of hotel facilities, especially in the circumstances of the COVID-19 pandemic, are supported by modern tools and technologies in the field of automation and robotization. According to the respondents, innovative technological solutions are appropriate in most areas of hotel operation. They are most often used to support processes in the field of hotel guest service by the key stages of the stay, i.e., booking, check-in, staying in the facility, as well as completion of the visit and check-out. Accordingly, hotel businesses use comprehensive management support systems known as Property Management Systems (PMS). This type of software is suitable for serving hotel guests for 93%-5*h, 98%-4*h, and 82%-3*h. In addition, respondents indicated that solutions in this area are used in the area of booking services integrated with the Online Travel Agency (OTA) module and global distribution systems (GDS) in all five- and four-star hotels (100%-5*h, 100%-4*h) as well as by 87% of three-star hotels. PMS also supports other tasks performed by reception staff in 94%-5*h, 99%-4*h, and sales channel management by all (100%) five-star hotels, 93%-4*h and 75%-3*h.

Another group of solutions supporting business processes with the use of modern technologies, which as part of the software are also based on the functions of the ERP system, are service management modules (100%-5*h, 78%-4*h, 75%-3*h), knowledge management, and group work. In addition, PMS and ERP systems enable the automation of the sales process and customer relationship management (CRM). Nevertheless, the widespread access to the Internet influenced the use of e-CRM, i.e., a system for managing relations via the Internet, in hotel enterprises. It enables significant automation of service processes and obtaining information about hotel guests. The research shows that the CRM solution is used by 96%-5*h, 80%-4*h, and 69%-3*h.

According to the respondents, business processes are also improved by automating the circulation of documents. The implementation of this type of system enables digital document management and the use of electronic archiving processes. Electronic document flow is carried out using integrated information management systems-Enterprise Content

Management (ECM). They are equipped with several domain subsystems that perform tasks in the field of document management (DMS), electronic document flow (workflow), and business process management (BPM). This type of solution was applied by 90%-5*h, 94%-4*h, and only 29%-3*h. An extension of the document management function is the content management system (CMS), which is a solution for editing the content posted on the hotel's website. It allows hotels to make modifications and perform tasks without knowing the software. The CMS system was implemented by 94%-5*h, 98%-4*h, and only 38%-3*h.

On the other hand, the following IT systems, Office Automation Systems (OAS) and Computer-Aided Administration (CAA) computer systems are used to support office work in all (100%) five- and four-star hotels and less than half (48%) of the three-star facilities. In the field of distribution channel management, hotel enterprises use Channel Manager (CM) systems, which also act as an online sales channel manager. The CM system facilitates the observation of prices of competitive offers in distribution channels. Online distribution channel management supports processes in 88%-5*h, 90%-4*h, and 55%-3*h. The counterpart of the CRM system in managing contacts with suppliers is the Supplier Relationship Management (SRM) software. Such a system enables hotel companies to optimize their supply chain to control the cost of products and services provided by external suppliers. SRM software was implemented by 94%-5*h, 91%-4*h, and 50%-3*h.

Process automation also applies to the area of finance and accounting, which supports work in the field of hotel bookkeeping. The conducted research shows that hotel enterprises in the indicated area most often use an ERP system with a financial and accounting module (92%-5*h, 86%-4*h, and 91%-3*h).

In the area of handling purchasing processes, hotels use the ERP or PMS system module to a lesser extent. This type of platform is used by 69%-5*h, 68%-4*h, and 40%-3*h. The ERP system is also used by hotel enterprises in the area of human resources management using the module for employee training, training management, knowledge management, and human capital management (50%-5*h, 64%-4*h, and 44%-3*h). For this analysis, according to the Euclidean distance used, it should be stated that the cut-off point in the areas of application of hotel autonomy tools is the level of 80 for identical areas (red line, see Figure 3b). The designated boundary in the bond distance made it possible to distinguish five different groups of areas in which business processes classified into two segments are supported.

Managing a hotel business, excluding key tasks related to guest service, also involves the implementation of business processes in other areas of its operation. As a consequence, hotels implement modern technologies and tools that empower, automate, and increase the efficiency of planning, service (guest service), and human resource management activities. Thus, based on the conducted research, it should be confirmed with the hypothesis that hotel facilities, through the processes of automation, robotization, and optimization, implement modern tools and technologies shaping a new model of market operation based on the autonomization of functions and tasks that create value for guests in the turbulent reality caused, among others, by the COVID-19 pandemic.

### 3.4. Motives for Implementing Autonomization Tools in Hotels

Nowadays, in the conditions of progressing globalization and the ongoing COVID-19 pandemic, almost every sector of the economy is subject to the influence of modern digital technologies, robotization, and automation. These types of tools implement processes in hotel companies based on precisely defined rules that are systematically repeated. Tasks are entrusted to robots which, using algorithms, rules, or instructions integrated with a cause-and-effect relationship, supervise the proceedings leading to the solution of the problem. For this reason, hotel facilities should not marginalize the impact of technology on individual areas of their operation.

In the current market conditions, an important motive that encourages business managers to implement autonomization tools has an economic and social character. In line

with its essence, the model of hotel operation is changing, which, thanks to digital transformation and automation, is based on providing the conditions for the implementation of a development strategy leading to satisfying the needs of hotel guests. As a result, in the next part of the research, the respondents identified the motives for the implementation of tools that streamline and automate the work of hotel staff.

Based on the data obtained from the respondents, to indicate the motives for implementing the above-mentioned automation and robotization tools, a cluster analysis was performed using the method of classifying average connections and the distance determined using the Euclidean metric (see Figure 4a,b).

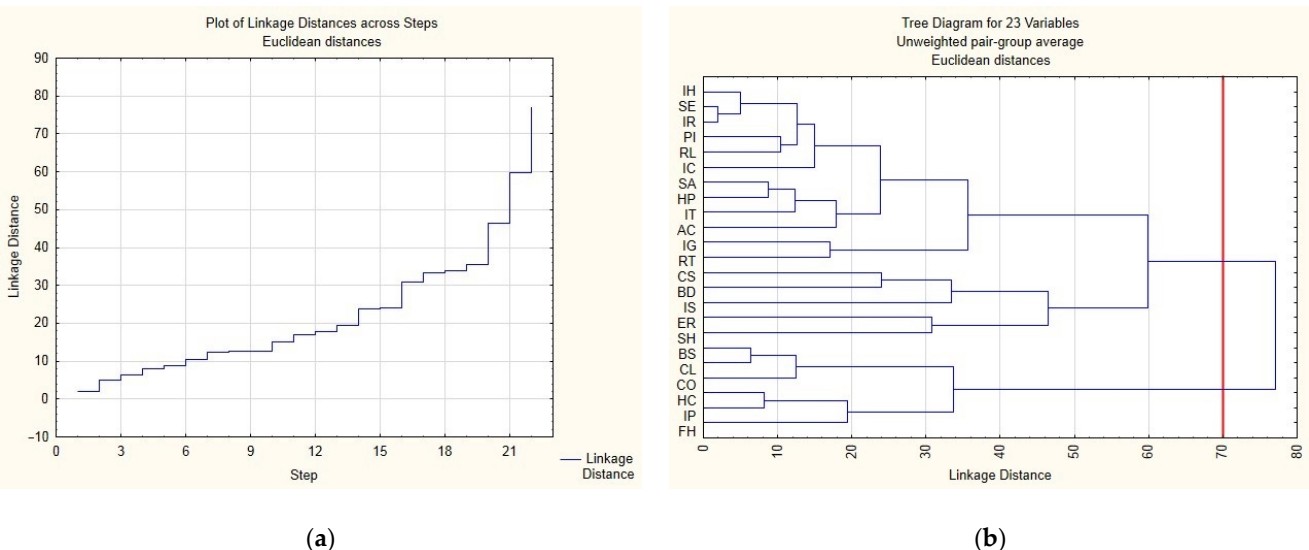

(**a**)                    (**b**)

**Figure 4.** Classification of motives for the autonomization of hotel enterprises: (**a**) The course of agglomeration for the main reasons for digitization, automation, and robotization of hotels; (**b**) The hierarchical diagram of the cluster analysis tree for the 23 rationales for hotel autonomy. Where: IH—improving the image of the hotel, CS—improving cooperation with suppliers, IS—improving data security, IG—improving the quality of hotel guest service, SE—improving service efficiency, SA—staff work automation, IR—increasing the range of services, BS—better service availability, CO—cost optimization, IR—increase in revenues, IC—improving customer relationships, HC—increase of the hotel's competitiveness, HP—improving the position on the market, AC—strengthening the competitive advantage, ER—employment reduction, PI—positive influence on the employment structure, FH—fulfillment of hotel norms and standards, RL—reduction in labor costs, SH—improving occupational safety and health, RT—limiting the response time to guests' wishes, BD—better use of data, IP—increase in employee productivity, CL—increase in customer loyalty. Source: Own study based on the Statistica 13.3 package.

According to the binding distance agglomeration diagram, the number of clusters equal to 3 was determined (see Figure 4a). The typification of the motives for the autonomy of hotel enterprises was stopped at the 20th stage of agglomeration, which is characterized by a distance of 46. Nevertheless, a significant increase in the bond distance was also recognized in the case of the extreme 22nd step with a distance of 88. According to the graphical interpretation of the agglomeration diagram, it should be noted that the fragmentation of the dendrogram should be performed after the defined last step (the longest vertical line, see Figure 4a).

On the other hand, the dendrogram presents the hierarchical structure of the analyzed variables, shaped as a result of the minimization of convergence between the classified themes of hotel autonomy. Based on the Euclidean distance criterion, the value of the cut-off point is located in the 70th degree, which determines the identical communities that categorize the motives of autonomy (red line, see Figure 4b).

The analysis of clusters for 23 motives of autonomy of hotel enterprises made it possible to group the analyzed circumstances into nine basic groups (see Figure 4b). The first set consists of the following reasons for the use of autonomization tools: increase in employee productivity (IP) and increase in hotel competitiveness (HC). The second group includes cost optimization (CO), increased customer loyalty (CL), and better service availability (BS). The third group consists of the improvement of occupational health and safety (SH) and reduction of employment (ER). The fourth summary includes improved data security (IS), better data use (BD), and improved cooperation with suppliers (CS). The fifth group is made up of improving the quality of hotel guest service (IG) and reducing the response time to buyers' requests (RT). The sixth group is represented by premises relating to the strengthening of the hotel's competitive advantage (AC). The seventh relationship is determined by the circumstances indicating an increase in the range of services (IT), improvement of the hotel's competitive position on the market (HP), and automation of staff work (SA). In the eighth cluster, improvement in customer relations (IC) was selected. The last set of autonomization motives includes reduction of labor costs (RL), positive impact on the employment structure (PI), increase in revenues (IR), improvement of service efficiency (SE), and fulfillment of hotel norms and standards (IH).

The research shows that the main reason for the autonomy of hotel enterprises is the need to enhance the efficiency of the service provision process (100%-5*h, 100%-4*h, and 86.2%-3*h). Moreover, the main motive for the implementation of this type of modern tool is to increase competitiveness (100%-5*h, 83.4%-4*h, and 74.6%-3*h). Nowadays, robotization and automation are an innovative form and model for the development of a hotel enterprise leading to its autonomization.

## 4. Discussion

The contemporary market conditions have obligated hotel companies to change the way they operate combined with immediate and unexpected automation, robotization, and digitization of implemented processes. In the conditions of the COVID-19 pandemic caused by the SARS-CoV-2 virus, it should be noted that the technological environment of the organization is, to a large extent, the most dynamic element of the macro-environment. Intensive progress in the implementation of new technologies, which has so far been observed especially in production processes, is now also of key importance for enterprises from the hotel services sector. The significant degree of automation and robotization of selected processes in hotels is determined by the access to a fast and safe internet network with high capacity.

The subject of the study was hotel enterprises considered from the perspective of tourism market entities, classified in the category of tourism service providers. Their functioning results from a deliberately organized, economically independent, and separated in terms of technical, service, spatial, and legal resources, including tangible and intangible resources, organized to meet the needs of tourists in the sphere of providing hotel services through individually made strategic and tactical decisions by managers [28].

The fundamental structure of the accommodation base is the hotel. Defined as an entity that rents rooms and provides basic (accommodation, meals, drinks) and additional services (e.g., entertainment, recreational and other) by the standard defined by the requirements for various types of criteria established by the categorization system and classified into the category one-, two-, three-, four-, or five-stars [29].

For this research, it was assumed that an autonomous hotel provides tourists with a higher quality of services through the use of new technologies such as ICT/IoT (Internet of Things), ALI, VR, RBR, RPA, digitization, and new organization and management processes, aiming to achieve greater satisfaction with a stay in the facility and provides employees with a better workplace. The component that integrates the indicated elements is technology combined with an intelligent management system supported by qualified human resources, which determines the market success of the hotel [30].

The conducted research made it possible to distinguish tools and technologies considered strategic in the transformation and creation of an autonomous hotel model resulting from functioning in the conditions of the COVID-19 pandemic. The issue of process autonomization has not yet been undertaken and analyzed from the perspective of transforming the models of entities functioning in the hotel services market.

In studies on the development of hotel and catering services [31], the authors mainly point to the potential for the development of digital services and digital transformation in the functioning of entities. In addition, they identify the impact of innovative digital solutions in the hospitality sector. The research also focuses on the analysis of the current state and the prospects for the development of digital transformation from the perspective of Industry 4.0 in hotels and restaurants. On their basis, the authors defined the conditions necessary to carry out the digital transformation and reduce the risk of implementing innovative digital technologies.

The research carried out for this article shows that the level of automation and robotization of selected processes depends on access to the Internet because hotels use mainly fiber optic internet to implement integrated business processes. By connecting to the network, they have at their disposal various types of wireless technologies, most often such as Bluetooth and Beacon tools that communicate with mobile devices such as a notebook, smartphone, and netbook. The use of modern communication technologies contributes to creating a new dimension in relations with hotel guests. The internet is a source of multilateral communication, searching for information, concluding transactions as well as co-creating value. As a consequence, hotel companies more and more often establish relationships with the use of this type of tool. Thanks to them, they achieve tangible benefits resulting from cost reduction, acquiring new buyers of hotel services, and digital transformation, based on which they shape autonomous models of functioning in the hotel market.

Some authors consider the automation of business processes in independent hotels from the point of view of tools ensuring market competitive advantage [32]. The diagnosis made it possible to identify the main trends resulting from the use of PMS in the hotel industry. Moreover, the authors proposed a methodology for the implementation of this type of system.

Nevertheless, the conducted research justifies the use of only integrated PMS systems supporting the work of the reception, booking (OTA, GDS), sales, and marketing departments. It works by processing huge amounts of data, which are then used to improve operational and tactical performance. The data analysis performed by the system enables rational strategic decisions to be made. In addition, an important system supporting business processes implemented by hotel enterprises is ERP, including CRM, eCRM, ECM, DMS, BPM, CMS, OAS, CAA, CM, and SRM modules. It is used for resource and process planning in the following departments, including sales, finance, reception, booking, accounting, human resources, marketing and sales, procurement, catering, and technical production. The main task of the system is to support business processes related to enterprise management. These include activity monitoring, cost optimization, striving to increase efficiency, helping in gaining a competitive advantage, increasing market share, and maximizing profit.

An interesting research problem was undertaken in one study [33], which concerns the use of digitization solutions and technologies for monitoring corrupt practices and unethical behavior in the hotel sector. The authors carried out an analysis that shows that in developed countries with a high level of digitization, the possibility of corrupt practices is limited. Thus, they conclude that digitization is an international anti-corruption instrument used in the hotel services market.

Nevertheless, most of the contemporary studies concern the analysis of the literature on the subject in the field of automation and robotization and the use of artificial intelligence in healthcare or industry during the COVID-19 pandemic [34–37]. In the publications, the authors emphasize the arguments that indicate the implementation of tools related to au-

tomation and robotization due to the change in customer preferences, increasing knowledge of artificial intelligence, and increasing confidence of entrepreneurs in automation tools.

In turn, the conducted research identifies, in addition to the BPA, RPA, and ALI tools indicated in the above-mentioned publications, also RBR, LCO, TEM, iBPMS, and TEM. These types of technologies have been available to managing organizations for several years. At present, however, a breakthrough in their application should be sought, which is the consequence of (1) Above-average results of process automation in the industrial sector; (2) Above-average level of process automation that can be achieved by implementations of tools in the field of artificial intelligence; (3) Changes in the positioning of business processes carried out in enterprises; (4) Changes taking place in the environment of the enterprise (pandemic and economic crisis); (5) Creation of new autonomous models of enterprise functioning.

Based on the conducted research, it should be concluded that the autonomous business model is of key importance as it enables confronting the challenges posed by the environment. These include, first of all, the digital revolution, cost reduction, innovative methods of serving hotel guests, and modern forms in the management and development of enterprises. Autonomous models of hotel functioning effectively solve the indicated problems and contribute to their transformation from reactive entities into proactive leaders in process automation and robotization.

The results obtained during the research carried out for this study do not constitute grounds for direct comparison with the discussions of other authors. This type of reasoning should be seen in the shortage of studies analyzing the identical issues concerning the applied tools of autonomization in the conditions of the COVID-19 pandemic. In addition, the currently available literature on the subject describes the impact of process automation and robotization technologies on the improvement of production processes. Due to the specificity of services and their intangible nature, it is not possible to compare the above-mentioned tools for self-optimization with entities operating in the industrial sector. The proposal for the use of individual tools of autonomy indicated in the article is not universal due to the variability of the environment of hotel enterprises. Nevertheless, it can inspire their implementation in terms of improving the services provided in crises caused, inter alia, by the pandemic.

Future research in the field of autonomization should focus on issues related to hyper-automation resulting from the use of the eco-system of advanced tools in the field of intelligent technologies for automating business processes implemented in turbulent market conditions (e.g., economic crisis, pandemic crisis, economic downturn, and new market segments).

## 5. Conclusions

The pandemic crisis that has lasted since 2019 has put into question the possibility of running a hotel and catering business. At the same time, with the emergence of new waves of the SARS-CoV-2 coronavirus, hotel enterprises are struggling with the need to adapt to new operating conditions. They result, inter alia, from maintaining social distance, restriction of movement, an adaptation of activities to issued epidemiological recommendations, and the application of preventive measures aimed at minimizing the risk of infection. Due to the indicated requirements resulting from the COVID-19 pandemic, hotel enterprises have started to implement modern solutions in the field of automation and robotization of processes.

The study showed that hotel enterprises, through the processes of automation, robotization, and optimization, implement modern tools and technologies shaping a new model of market operation based on the autonomization of functions and tasks that create value for hotel guests in a turbulent reality caused, among others, by the COVID-19 pandemic. Thus, it was possible to confirm the hypothesis. Nevertheless, the level of automation and robotization, depending on the hotel category, depends on access to a specific type of internet network and the mobile devices used.

Automation and robotization processes contribute to the change or modification of the functioning model of hotel entities from digital to autonomous ones. They are implemented using the following tools, RPA, LCO, iBPMS, ChV, OCR, and BPA, which are implemented in the following areas of the company's operation in the fields of: (1) service management, content, customer relations, and online sales channels; (2) supporting the work of reception; (3) handling sales processes, hotel guests; (4) automatic document circulation. Autonomization tools are implemented in the above-mentioned areas of operation of hotel enterprises due to the need to strengthen the efficiency of the service provision process, which enables them to increase their competitiveness and innovativeness in the face of economic and pandemic crises.

The result of the implementation of automation and robotization tools for business processes in hotels is to increase the efficiency of their operation by increasing the potential, which translates into time savings in the tasks performed in the above-mentioned areas of activity. In an autonomous hotel enterprise, the use of modern technologies of digitization, automation, and robotization of processes improves communication between employees and multiplies the efficiency of their work. In addition, it enables a more beneficial use of material and non-material resources at the disposal of the enterprise. The implementation of such improvements results in a reduction in the costs of running a business. Thus, the use of RPA, LCO, iBPMS, ChV, ALI, OCR, BPA tools in supporting processes may contribute to achieving a competitive advantage and innovativeness of hotels, because they support operational, tactical, and strategic activities, especially in the turbulent environment of entities currently shaped by the COVID-19 pandemic.

Autonomization of hotel enterprises as a result of using the indicated solutions will bring significant benefits from the perspective of changes in the employment structure and minimization of operating costs. The use of automation and robotization to perform repetitive processes also enables hotel managers to: (1) Shift employees from positions requiring the performance of routine tasks to strategic activities affecting the achievement of competitive advantage; (2) Elimination of imperfections usually resulting from the employee's lack of involvement in the tasks performed; (3) Speeding up the execution of tasks thanks to the repeatability of the robot's actions; (4) Standardization of processes by developing detailed procedures and description of the workflow in automated processes; (5) The relatively short-term implementation phase, which ensures a quick return on investment; (6) Changing the hotel business model from digital to autonomous; (7) Full control of the tasks performed in the hotel; (8) Use of ICT infrastructure and owned mobile devices; (9) Reduction of the number of errors in the hotel guest service process; (10) Reduction of the number of errors, i.e., increasing the efficiency of processes.

The indicated conclusions were supported by proprietary research conducted among entities operating on the international market of hotel services. The study showed an urgent need to implement technologies in the field of automation and robotization, which enable the creation of an effective model of functioning with the use of the indicated tools in the conditions of the transforming external environment.

**Funding:** This research received no external funding.

**Institutional Review Board Statement:** Not applicable.

**Informed Consent Statement:** Not applicable.

**Data Availability Statement:** The data presented in this study are available on request from the corresponding author.

**Conflicts of Interest:** The author declares no conflict of interest.

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
