# Peer review of "Autonomous Enterprise as a Model of Hotel Operation in the Aftermath of the COVID-19 Pandemic"

_sustainability, doi:10.3390/su14010097_

Round 1
Reviewer 1 Report
The article aims to identify modern technological solutions in the field of automation and robotization of business processes that change the way hotel enterprises operate in the face of the global Covid-19 pandemic. Original, and well-written paper. Methodology is unique. Results are useful.
Please separate the Introduction part: Introduction and independent Literature review.
Author Response
Dear Reviewer, I would like to thank the Reviewer for the effort to read the article entitled "Autonomous Enterprise as a Model of Hotel Operation in the Aftermath of the Covid-19 Pandemic", kind assessment and constructive attention that allowed the content to be refined. After reading the review of the article, the following comments and suggestions were taken into account: 1. The introduction has been separated from the literature review (L47-78) Once again, I would like to thank the Reviewer for the effort put in preparing the review, as well as thorough attention and rightly indicated imperfections. The comments included in the review will certainly also be helpful in the future in the precise editing of articles. I wish you a Merry Christmas and a Happy New Year. Yours faithfully author
Reviewer 2 Report
Dear author,
The topic of the article is very actual and it helps to understand better the need of technology adoption in tourism. I find the article well structured, with many useful information for the stakeholder. Please find below my suggestions for improving the manuscript:
- please add in the abstract the country where the research was conducted
- add the information about the country where are located the hotels included in the study at the end of Introduction section (L130-L1135)
- write the 4 questions from the Materials and Methods in a more attractive way (include them in a table or write them as a list) - L146-L151
- please expand the discussion section with more info related to the results of the present paper
- add the limits of the study at the Discussion section
All the best!
Author Response
Dear Reviewer,
I would like to thank the Reviewer for the effort to read the article entitled "Autonomous Enterprise as a Model of Hotel Operation in the Aftermath of the Covid-19 Pandemic", kind assessment, constructive comments, and conclusions that allowed for the improvement of the content. After carefully reading the review of the article, the following comments and suggestions were taken into account:
- The country in which the research was carried out was added (following the suggestions of Reviewer 1, the paragraph was moved to the introduction, the country supplement was entered in L68-69).
- Research questions are included in Table 1 (L180-181).
- The boundaries of the research field were supplemented according to the subjective, objective, and aspect criteria (L669-687).
- Reference was made in the discussion to information obtained during the study (L701-713, L720-732, L747-763).
I would like to thank the Reviewer for the effort put in preparing the review, as well as insightful comments and rightly indicated imperfections regarding the presented issues. The comments included in the reviews will certainly also be helpful in the future in the precise editing of articles.
I wish you a Merry Christmas and a Happy New Year.
Yours faithfully
author

Reviewer 3 Report
- Update the reference list
- Better make clear the managerial implications
- Better make clear conclusions
Author Response
Dear Reviewer,
I would like to thank the Reviewer for the effort to read the article entitled "Autonomous Enterprise as a Model of Hotel Operation in the Aftermath of the Covid-19 Pandemic", kind assessment, constructive comments, and conclusions that allowed for the improvement of the content. After carefully reading the review of the article, the following comments and suggestions were taken into account:
- The publication was updated from the reference list
- Management implications were indicated (L822-835)
- Efforts were made to present the main conclusions more precisely (L809-L821)
I would like to thank the Reviewer for the effort put in preparing the review, as well as insightful comments and rightly indicated imperfections regarding the presented issues. The comments included in the reviews will certainly also be helpful in the future in the precise editing of articles.
I wish you a Merry Christmas and a Happy New Year.
Yours faithfully
author
